# Modulatory Effects of Hydatid Cyst Fluid on a Mouse Model of Experimental Autoimmune Encephalomyelitis

**DOI:** 10.3390/vetsci11010034

**Published:** 2024-01-15

**Authors:** Maryam Hajizadeh, Aynaz Jabbari, Adel Spotin, Seyyed Sina Hejazian, Tahereh Mikaeili Galeh, Hadi Hassannia, Maryam Sahlolbei, Abdol Sattar Pagheh, Ehsan Ahmadpour

**Affiliations:** 1Infectious and Tropical Disease Research Center, Tabriz University of Medical Sciences, Tabriz 51656-65811, Iran; 2Immunology Research Center, Tabriz University of Medical Sciences, Tabriz 51656-65811, Iransina.hej95@gmail.com (S.S.H.);; 3Neurosciences Research Center, Tabriz University of Medical Sciences, Tabriz 51656-65811, Iran; 4Department of Basic Medical Sciences, Khoy University of Medical Sciences, Khoy 53464-58167, Iran; 5Immunogenetic Research Center, Faculty of Medicine and Amol Faculty of Paramedical Sciences, Mazandaran, University of Medical Sciences, Sari 48175-866, Iran; 6Infectious Diseases Research Center, Birjand University of Medical Sciences, Birjand 14619-65381, Iran

**Keywords:** experimental autoimmune encephalomyelitis, multiple sclerosis, helminth immunotherapy, cystic echinococcosis, hydatid cyst

## Abstract

**Simple Summary:**

Multiple sclerosis (MS) is an autoimmune disease characterized by inflammation and demyelination of the central nervous system. The hygiene hypothesis suggests that a decrease in exposure to parasites may contribute to an increase in MS incidence. We hypothesized that cystic echinococcosis (CE), which is caused by the helminth *Echinococcus granulosus*, could potentially reduce the severity of MS. To investigate this, we used mice with experimental autoimmune encephalomyelitis (EAE), a model of MS. The mice were injected with hydatid cyst fluid (HCF) from *E. granulosus* cysts. The HCF shifted the immune response from a pro-inflammatory Th1 to an anti-inflammatory Th2, resulting in a decrease in IFN-γ and an increase in IL-4. As a result, the HCF delayed the onset of EAE and reduced its severity. This led to the prevention of weight loss and a decrease in inflammation and demyelination in the spinal cord.

**Abstract:**

The reduced burden of helminth parasites in industrialized countries is probably one of the reasons for the increased prevalence of autoimmune disorders such as multiple sclerosis (MS). The current study aimed to evaluate the potential preventive effects of hydatid cyst fluid (HCF) on the disease severity in an EAE mouse model of MS. EAE-induced mice were treated with HCF before and after EAE induction. An RT-PCR-based evaluation of IFN-γ, IL-1β, TNF, T-bet, IL-4, GATA3, IL-17, RoRγ, TGF-β, and FOXP3 expression levels in splenocytes and an ELISA-based analysis of IFN-γ and IL-4 levels in cell culture supernatant of splenocytes were performed. Histopathological examinations of mice during the study were also conducted. The expression levels of T-bet, IL-4, GATA3, TGF-β, and FOXP3 in EAE + HCF mice were significantly higher compared to EAE + PBS mice. In the EAE + HCF group, the expression levels of IFN-γ, IL-1β, and TNF were significantly lower than in the EAE + PBS group. The histopathological results showed significantly reduced inflammation and demyelination in EAE + HCF mice compared to EAE + PBS mice. Our study provides proof-of-concept in the EAE mouse model of MS that helminth-derived products such as HCF have a potential prophylactic effect on MS development and present a novel potential therapeutic strategy.

## 1. Introduction

Widespread industrialization in the last century and subsequent improvements in sanitation have significantly reduced the prevalence of helminth parasites [1]. According to the hygiene hypothesis, the decreased burden of helminth parasites in industrialized countries is one reason for the increased prevalence of allergies and asthma in these nations [2,3]. This aligns with the well-established fact that helminth parasites induce a state of reduced host immune response, subsequently lowering the incidence of inflammatory diseases like autoimmune disorders in infected patients [2,3]. Industrialized countries are currently experiencing a rise in the incidence of multiple sclerosis (MS), which is likely attributed to a decline in parasitic infections [4,5].

MS is an autoimmune disease of the central nervous system (CNS) and is the most frequent non-traumatic neurological cause of disability in youth. It is characterized by lesions in the brain that lead to a variety of neurologic symptoms, including paralysis, vision loss, and coordination loss [6,7,8,9]. Multiple mechanisms are involved in the pathogenesis of this inflammatory, demyelinating, and neurodegenerative disease, and they are mostly recruited by autoreactive T helper cells, specifically the Th1 and Th17 subsets [10]. Entrance and activation of these cells in the CNS lead to the formation of inflammatory lesions, resulting in myelin loss, demolition of oligodendrocytes, and axonal injury [11,12]. Despite tremendous efforts in the treatment of MS patients, the currently available drugs are mostly effective in the relapsing-remitting form of the disease (RRMS) and reduce the debilitating symptoms [13,14]. Thus, there is always a need for novel therapeutics. It is worth mentioning that immunomodulation is the most common action ofdrugs used to treat MS patients [15].

Cystic echinococcosis (CE) or hydatidosis is a major neglected zoonotic disease in many parts of the world and is caused by the larval stages of cestodes from the genus Echinococcus [16,17]. Human beings are accidental or aberrant intermediate hosts and become infected through direct contact with infected final hosts or by ingesting food, water, or soil contaminated with embryonated eggs of the parasite [18,19,20]. The formation of cystic lesions in the liver, lung, kidney, or other parts of the body is characteristic of CE [16,18]. During the process of pathogenesis, oncospheres released from parasite eggs penetrate the human gut wall and enter the blood or lymph circulation, which can carry them to distant organs where they have the potential to develop into hydatids containing protoscoleces (PSCs) within months to years [21]. Hydatids are spheroidal, unilocular fluid-filled structures enveloped by a thin layer of parasite cells called the germinal layer (GL). The GL itself is surrounded by a much thicker cellular layer called the laminated layer (LL). Within the hydatids are brood capsules, which potentially contain PSCs. Some hydatids also contain self-resembling components known as daughter hydatids [22].

It is well known that helminths induce immunosuppressive changes to prolong their survival within hosts, including Th2 response enhancement, increased Treg numbers, and the expansion of myeloid-derived suppressor cells [23,24]. Previous studies have shown that helminth infection reduces the severity of experimental autoimmune encephalomyelitis (EAE), a mouse model of multiple sclerosis (MS) [23,24]. As a helminth infection, not only does cystic echinococcosis (CE) protect itself against local host-derived effectors by deploying a physical shield consisting of the laminated layer (LL) and adventitia [22,25,26], but it also induces a shift toward a Th2-based response, which has immunoregulatory effects. However, Th1-related parameters are also found to be elevated in infected patients [27,28,29]. We hypothesized that CE infection in the EAE mouse model of MS may reduce inflammatory responses and alleviate disease severity. Therefore, the current study aimed to examine this hypothesis by evaluating the potential preventive effects of hydatid cyst fluid (HCF) on disease severity in an EAE mouse model of MS.

## 2. Materials and Methods

### 2.1. Animals 

A total of 24 adult female C57BL/6 mice, aged 9–10 weeks and weighing 21–24 g, were obtained from the Pasteur Institute of Iran (Tehran, Iran). The mice were transferred to the Parasitology Research Center at Tabriz University of Medical Sciences, where they were housed in standard polycarbonate cages under controlled conditions, including a temperature of 23 ± 1 °C and a 12 h light/12 h dark cycle. The study protocols were approved by the Ethics Committee and Animal Care and Use Committee of Tabriz University of Medical Sciences (Code number: IR.TBZMED.REC.1399.474). 

### 2.2. Preparation of HCF

In this study, infected sheep livers were procured regularly from a slaughterhouse in East Azerbaijan province, Northwest Iran. Only fertile cysts containing protoscoleces (PSCs) were selected, while calcified or penetrated cysts were excluded. Initially, a sheep host was utilized to obtain cyst fluid for antigen preparation, as cysts in cow samples were found to be infertile and devoid of protoscoleces, affecting the study’s consistency. To ensure uniformity, the host tissue sample chosen for collection was heavily infected with numerous cysts. Prior to hydatid cyst fluid aspiration, the cyst surface was cleansed with 75% ethanol. Subsequently, aspiration of hydatid cyst fluid (HCF) was performed using a sterile 20 mL syringe and collected in 50 mL sterile Falcon tubes. Collected HCF tubes were centrifuged at 800 g for 15 min. PSC viability was confirmed by observing the death of flame cells stained with 0.1% eosin under a light microscope. Following molecular confirmation of the genotype (results indicated that the predominant genotype in the Azerbaijan region of Iran is G1), the hydatid cyst fluid was collected and stored at −20 °C. To maintain sample consistency and integrity, the protein concentration in the collected hydatid cyst fluid was determined using the Bradford method. Finally, the HCF was filtered through a 0.22 μm mesh size and stored for future use.

### 2.3. EAE Induction and Intervention

The EAE induction in mice was performed using 200 μg of myelin oligodendrocyte glycoprotein peptide (MOG 35–55; SICBD, Karaj, Iran) emulsified with 500 μg of heat-killed **Mycobacterium tuberculosis** in complete Freund’s adjuvant (CFA; Sigma Co., Burlington, MA, USA). Each component was dissolved in 50 μL of PBS. The MOG 35–55 + CFA emulsion was subcutaneously injected into both hind flanks of the mice. Additionally, 100 μL of PBS containing 300 ng of pertussis toxin (PTx) was intraperitoneally injected at two distinct times (time of induction and 48 h after that) to assist with the immunization of the mice [30].

### 2.4. Experimental Groups and Treatments

The mice were randomly divided into three groups (n = 6) including (1) a healthy control group with no specific intervention (C), (2) an EAE-induced group with no specific intervention (EAE + PBS), and (3) an EAE-induced group treated with HCF (EAE + HCF). The C and EAE + PBS mice received only intraperitoneal PBS (100 μL). The EAE + HCF mice were treated with 50 μg/mL of HCF 2, 4, and 6 days before and after EAE induction (Figure 1). The following criteria, based on a five-point scale, were used to determine the clinical score of mice based on their symptoms: (0) no symptoms, considered healthy; (1) loss of tail tone; (2) paresis of hind limb; (3) paralysis of hind limb; (4) paralysis of both hind and forelimbs; and (5) near death or death [31]. Furthermore, the body weight was intermittently measured daily.

### 2.5. Histopathological Examination and Cell Preparation

After being anesthetized with an intraperitoneal injection of 50 mg/kg ketamine and 5 mg/kg xylazine in a 20 μL volume (Alfasan, Woerden, Netherlands), mice were sacrificed on the 21st day following EAE induction. The lumbar spinal cord was extracted and fixed in 4% paraformaldehyde (pH 7.2). The samples were subsequently divided into four levels and embedded in paraffin. They were then sectioned into 10 micrometer-thick slices. The sections were stained using both Hematoxylin and Eosin (H&E) to assess inflammatory cell infiltration and Luxol Fast Blue (LFB) to evaluate demyelination. Slides were analyzed by a skilled pathologist, blinded to the study, using a light microscope [31].

### 2.6. Isolation of Splenocytes

After scarification of the mice, their spleens were extracted and prepared for further analysis. Briefly, splenocytes were collected from the spleens and prepared for molecular studies using sterile Roswell Park Memorial Institute (RPMI)-1640 medium. Subsequently, the cell suspension was filtered through a mesh and washed three times with sterile RPMI medium to address cell clumping.

### 2.7. Quantitative Real-Time Reverse Transcriptase–Polymerase Chain Reaction (qRT-PCR)

The expression levels of genes and transcription factors associated with immune cells, including Th1 (INF-γ, IL-1β, TNF, and T-bet), Th2 (IL-4 and GATA3), Th17 (IL-17 and RoRγt), and Treg (TGF-β and FOXP3), were measured in the isolated splenocytes using qRT-PCR. Total RNA (mRNA and miRNA) was extracted using the TRIzol^®^ reagent (GeneAll, Biotechnology, Seoul, Republic of Korea). RNA quality and quantity were determined using gel electrophoresis and a NanoDrop spectrophotometer (ND1000 Technologies, Wilmington, DE, USA), respectively. Complementary DNA (cDNA) synthesis was performed according to the manufacturer’s instructions (Thermo Fisher Scientific, Inc., Waltham, MA, USA). The qRT-PCR reaction mixture, with a final volume of 20 μL (achieved with distilled water), consisted of 1 μL cDNA, 10 μL of 2X SYBR-Green master mix (Biofact, Daejeon, Republic of Korea), and 2 μL of both reverse and forward primers. The temperature cycle included 40 consecutive cycles of 95 °C for 15 s, 55 °C for 30 s, and 72 °C for 30 s. β-actin was used as the internal control for quantifying and relative gene expression using the 2^−ΔΔCt^ method.

### 2.8. Enzyme-Linked Immunosorbent Assay (ELISA)

Splenocytes obtained from mice were cultured in 24-well plates and treated with MOG35-55. After a 72 h incubation, supernatants were collected to assess IFN-γ and IL-4 cytokines using ELISA kits (Mabtech, Nacka, Sweden), according to the manufacturer’s protocols. Briefly, 100 microliters of monoclonal antibodies, diluted to 2 micrograms per milliliter in phosphate-buffered saline (PBS), were dispensed into each well of the ELISA plate and incubated at 4 °C overnight. The following day, the plate was washed with PBS supplemented with 0.05% Tween 20 (PBST), and each well was blocked by adding 200 microliters of 0.1% bovine serum albumin in the same wash solution and incubated at room temperature for 90 min. Afterward, 100 microliters of either the standard solution or the supernatant were added to each well, followed by a 2 h incubation at room temperature. Post-incubation, the plate was washed again, and 100 microliters of the detection antibody were added to each well, with a subsequent incubation for one hour. Next, 100 microliters of Streptavidin-HRP conjugate were added to each well and allowed to incubate for 30 min at room temperature. Following another washing step, 100 microliters of TMB substrate were added to each well, and the plate was shielded from light for 15 min at room temperature. The enzymatic reaction was stopped by adding 50 microliters of 2 N sulfuric acid, and the optical density was quantified at 450 nm using an ELISA microplate reader (BP800 Microplate Reader, Biohit Plc, Helsinki, Finland).

### 2.9. Statistical Analysis

All qualitative and quantitative variables were reported using frequency (percent) and mean ± standard deviation (SD), respectively. One-way ANOVA analysis was applied to compare the studied quantitative parameters among the three mice groups. Subsequently, the Tukey post hoc test was used to determine which pair of groups caused a significant difference among the three mice groups. All statistical analyses were performed using Graph Pad Prism software version 7 at a significance level of 0.05. All experiments were performed in triplicate, and the data were expressed as the mean ± standard deviation (SD).

## 3. Results

### 3.1. Average Clinical Scores of Mice

Evaluation of the average clinical scores of the mice demonstrated that the administration of HCF exerted protective effects in the EAE-induced mice. As depicted in Figure 2, this protective effect manifested in two distinct aspects: a delay in the onset of EAE symptoms and a reduction in the maximum severity of the disease. Moreover, the mean age at disease onset was 12 ± 2 days, and the mean clinical score was 5 ± 1, with a 100% incidence in the EAE-induced mice.

### 3.2. Weight Alterations of Mice

The weight alterations of the mice in each group are displayed in Figure 2. The mice in the C group and the EAE + HCF group gained weight during the study period. However, the EAE + PBS group not only failed to gain weight but also lost weight slightly.

### 3.3. Histopathological Findings

The results of histopathological examination of the spinal cord slices stained with H&E and LFB revealed that the immune cell infiltration and demyelination rates in the EAE + HCF mice were significantly lower than those in the EAE + PBS group (Figure 3).

### 3.4. Results of Cytokine Assay Based on RT-PCR

The RT-PCR technique was employed to assess the expression levels of cytokine-related genes. The expression levels of INF-γ, IL-1β, and TNF genes were significantly decreased in EAE + HCF mice compared to the EAE + PBS group (Figure 4), indicating a decreased Th1-mediated immune response. No statistically significant difference was observed between the experimental groups regarding the two Th17-related proteins, IL-17 and RoRγ (*p* > 0.05). Additionally, the expression levels of T-bet (a Th1-related transcription factor), IL-4, GATA3 (Th2-related proteins), FOXP3, and TGF-β (Treg-related proteins) genes in the EAE + HCF mice were significantly higher compared to the EAE + PBS mice (*p* < 0.001) (Figure 4).

### 3.5. Evaluation of Cytokines Based on ELISA

The levels of IFN-γ and IL-4 cytokines were assayed in the supernatant of cultured splenocytes using ELISA (Figure 5). The level of IFN-γ was the highest among the EAE + PBS mice, which shows a decreased Th2 response. On the other hand, the level of IFN-γ was significantly lower in the EAE + HCF mice compared to the EAE + PBS mice (*p* < 0.01). The level of IL-4 was the lowest among the EAE + PBS mice, which indicates an increased Th1 immune response. However, no significant differences were observed regarding the IL-4 level between the experimental groups (*p* > 0.05) (Figure 5).

## 4. Discussion

CE infection is accompanied by a shifting immune response from Th1 to Th2, which is favorable for immune response suppression [32]. On the other hand, MS is a progressive and inflammatory disease that leads to myelin destruction of nerves in the CNS through a Th1-mediated immune response [33]. Currently, the routine therapeutic strategy in MS patients involves using drugs like dimethyl fumarate, which shift the immune response from Th1 to Th2. Multiple studies have demonstrated the efficacy of these agents [34,35,36,37,38]. Given the association of helminth infection with enhanced Th2-mediated immune response, we evaluated the effects of HCF on disease severity in an EAE mouse model of MS. Our results indicated a dominant shift in the inflammatory response from Th1 to Th2, as well as a significant reduction in EAE symptoms in the treated mice.

Although this study was conducted to evaluate the effects of products obtained from hydatid cysts on MS disease, several other studies have investigated the efficacy of helminth-induced immunotherapy (HIT) for MS, with most of them showing promising results. One study found that infection with *Trichinella pseudospiralis* in EAE mice led to reduced levels of IL-1β, TNF, and IL-17 in both their spinal cord and splenocytes [39]. In another study, the infection of EAE mice with *Heligmosomoides polygyrus* resulted in a decrease in the level of IFN-γ in their CNS [40]. More recently, Mariki et al. conducted a similar study using antigen B from hydatid cyst fluid [41]. The findings of the study revealed that antigen B of hydatid cyst fluid can modulate the immune response and improve nerve function in a rat model of MS. This was evidenced by increased nerve conduction and anti-inflammatory IL-10 levels. These effects suggest that antigen B may rebalance the immune response towards anti-inflammatory processes and promote myelin sheath repair, which is a critical factor in MS treatment. Hydatid cyst fluid was used in the present study due to the presence of numerous other antigens in the complete fluid of the hydatid cyst, which could potentially impact the process being investigated. In our study, to evaluate the activity of Th1 cells, which is usually increased in MS disease, the level of IFN-γ in the cell culture supernatant of splenocytes and expression levels of IFN-γ, IL-1β, TNF, and T-bet in splenocytes was measured. The level of IFN-γ in the EAE + HCF mice was significantly lower than in the EAE + PBS mice and close to that of the C mice, indicating a decreased Th1-mediated immune response. The expression levels of IL-1β and TNF were also reduced in the EAE + HCF mice compared to the EAE + PBS mice, further supporting the evidence of decreased Th1-mediated immune response in the EAE + HCF mice. However, the expression level of T-bet in the EAE + HCF mice was higher than in the EAE + PBS mice, which contradicts the previously mentioned findings of this study and other studies [33], as the elevated level of T-bet favors a shift toward Th1-mediated immunity. Many studies have reported that helminth infection with different helminths in EAE mice decreases Th1 cells in both the CNS [42,43,44,45] and peripheral circulation [39,43,46,47,48].

To assess the activity of Th2 cells, which is typically decreased in MS [33], we evaluated the level of IL-4 in the cell culture supernatant of splenocytes and the expression levels of IL-4 and GATA3 in splenocytes. Both parameters were higher in the EAE + HCF mice compared to the EAE + PBS mice, indicating an immune tendency towards a Th2-mediated response. While several studies have shown an increase in Th2 levels in the CNS [42,43] and periphery [42,44,47] of EAE mice after HIT, Reyes et al. (2011) reported no significant difference in the number of Th2 cells in EAE mice with Taeniacrassiceps infection [45].

We observed increased expression levels of TGF-β and FOXP3 in the EAE + HCF group compared to the EAE + PBS group, suggesting enhanced Treg cell activity and immune suppression. Previous studies have shown that HIT induction in EAE mice upregulates Treg-related cytokines [39,40,41,42,44,45,47]. Conversely, ROR-γ and IL-17 expression levels decreased in the EAE + HCF mice, indicating a shift in Th17 cell activity, which has been recently implicated in MS [49]. However, these changes were not statistically significant. Past studies have yielded conflicting results on this issue, with some reporting a decrease in Th17-related cytokines after HIT [39,43,44,50] and others reporting the opposite [40,45].

In addition to molecular studies, histopathological examination in our study revealed significantly less spinal cord injury in the EAE + HCF mice compared with the EAE + PBS group, which aligns with findings from studies using other helminths for HIT [40,42,44,45]. The protective effects of HCF against EAE were also evident in the overall clinical scores of mice, which were lower in the EAE + HCF group compared with the EAE + PBS group. Moreover, the up-regulated weight alterations observed in the EAE + HCF mice relative to the EAE + PBS mice during the study further indicate the protective role of HCF against EAE. Research has shown that drugs that shift the immune response from Th1 to Th2 reduce the severity and relapse episodes of multiple sclerosis (MS), leading to improved prognosis in patients [23]. In a review study on HIT in EAE mice, 20 out of 23 studies reported a prophylactic or alleviating effect of helminth-derived products on EAE [51]. In another study, the use of **Trichinella pseudospiralis** in mice prior to EAE induction resulted in a reduction in disease severity and a delay in the onset of symptoms [39]. Studies evaluating the efficacy of helminth infection therapy (HIT) in multiple sclerosis (MS) treatment are not limited to animal studies. Two prospective cohort studies on MS patients infected with various helminths demonstrated that these patients experienced fewer episodes of relapse, had positive radiologic findings, and exhibited increased and decreased levels of TGF-β and IFN-γ, respectively, compared to uninfected patients [52,53]. Interestingly, four patients in one of the studies were treated with anti-helminth agents, which led to increased episodes of relapse, radiologic findings, levels of IFN-γ and cells producing IL-12, and decreased levels of TGF-β and cells producing IL-10 [52].

One limitation of our study was the short follow-up period for intervened mice, which may not have been sufficient to observe the maximum severity of EAE. Moreover, since this was an animal study, the generalization of its results to MS patients is limited. Additionally, the study’s focus on prophylaxis limited the investigation of the therapeutic effects of the studied agents, as MS patients are typically symptomatic at the initiation of therapy. Another limitation is that the study did not examine the therapeutic potential of hydatid cyst fluid in established MS. It primarily focused on the preventive effects of HCF, specifically its ability to delay or prevent the onset of MS. Future studies should investigate the therapeutic potential of HCF for established MS to provide a more comprehensive understanding of its possible benefits.

## 5. Conclusions

To conclude, our study provides proof-of-concept in the EAE mouse model of MS that helminth-derived products such as HCF have a potential prophylactic effect on MS development and present a novel potential therapeutic strategy for MS patients. These results enhance the growing body of evidence that products obtained from helminth parasites are an inimitable resource for developing anti-inflammatory drugs. Accordingly, more studies are required to find out the exact beneficial aspects of helminth parasites in MS patients. 

## Figures and Tables

**Figure 1 vetsci-11-00034-f001:**
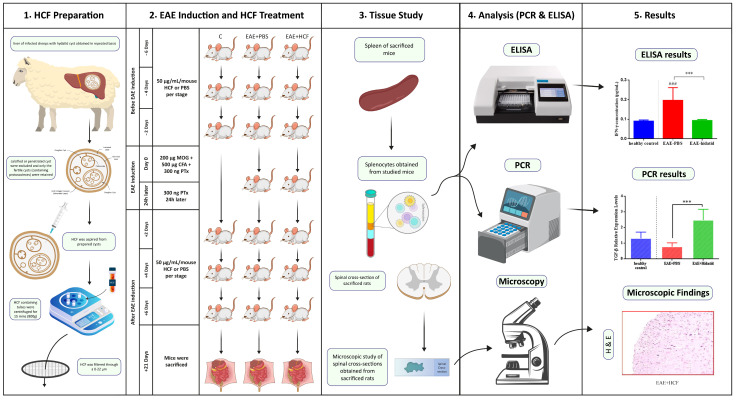
Summary of the experimental design. The *p*-value was determined by performing one-way ANOVA followed by Tukey’s test (*** *p* < 0.001) and (*^###^ p* < 0.001).

**Figure 2 vetsci-11-00034-f002:**
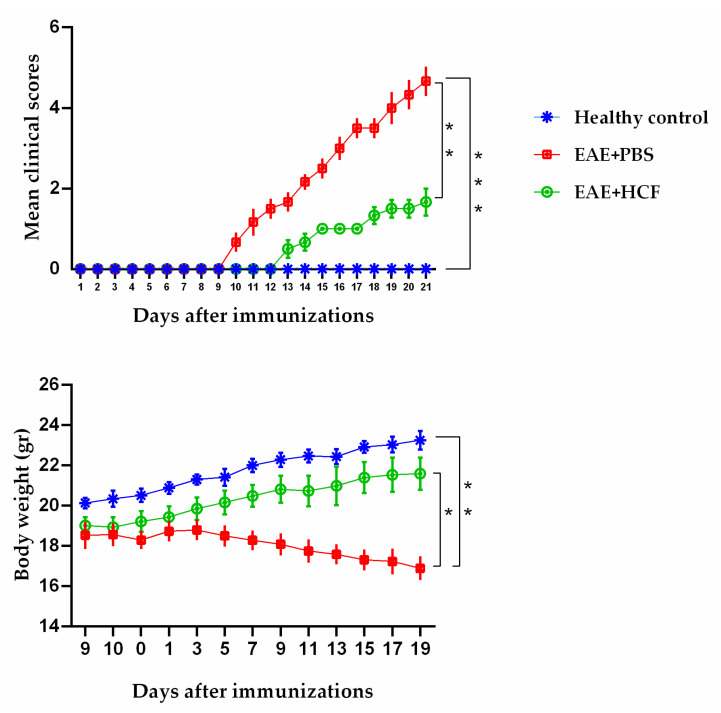
Weight alterations and average clinical score of mice during the study. Results are shown as mean ± SD (* *p* < 0.05, ** *p* < 0.01, *** *p* < 0.001). Six mice were randomly selected and examined from each group.

**Figure 3 vetsci-11-00034-f003:**
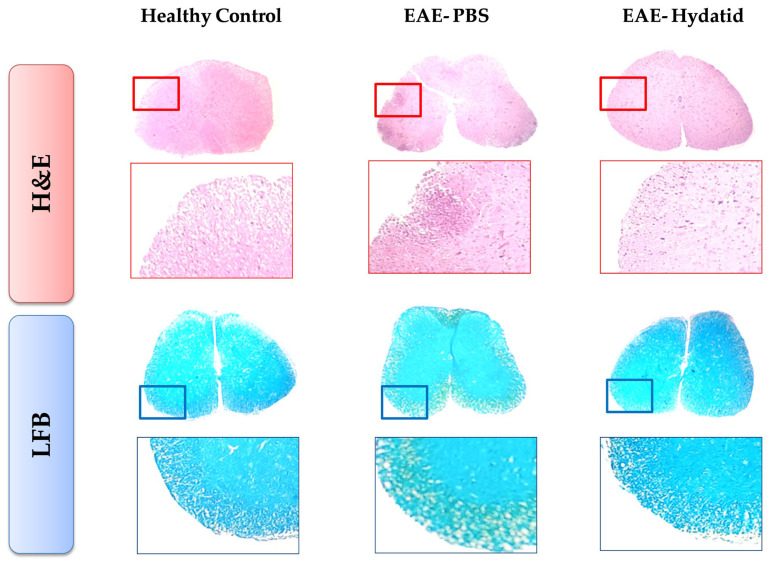
Demyelination and inflammation in the spinal cord. H&E-stained spinal cord sections of EAE treatment by PBS; EAE treatment by hydatid exhibited signs of reduced inflammation compared to that in the PBS group. Luxol Fast blue staining (LFB) in spinal cord sections of EAE treatment by PBS and EAE treatment by hydatid revealed signs of demyelination indicated by white areas in the nerve tissue.

**Figure 4 vetsci-11-00034-f004:**
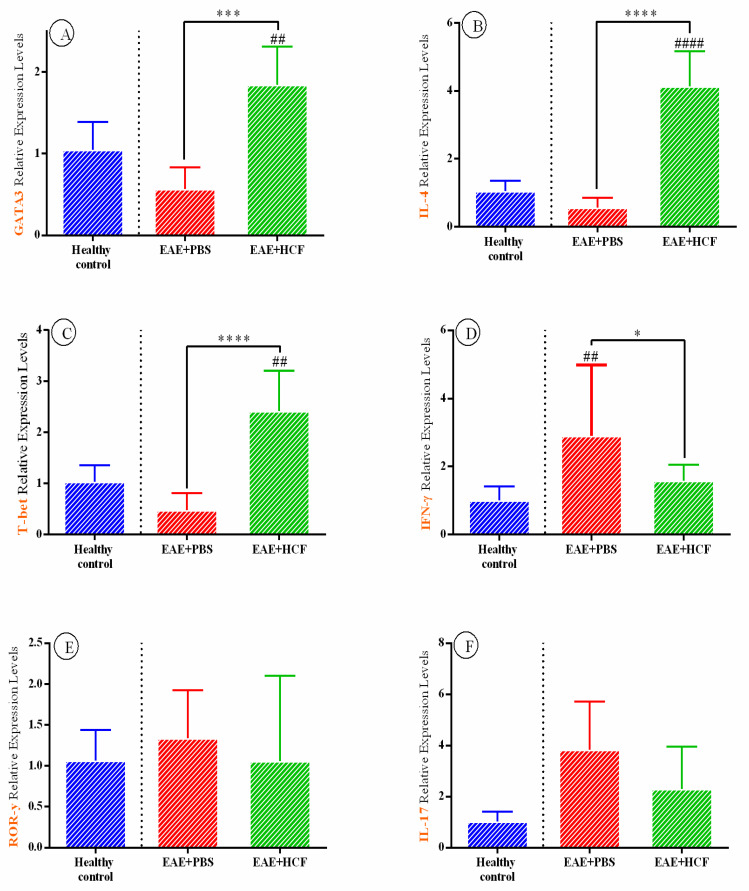
Gene expression levels of GATA3 (**A**), IL-4 (**B**), T-bet (**C**), IFN-γ (**D**), ROR-γ (**E**), IL-17 (**F**), FOXP3 (**G**), TGF-β (**H**), IL-1β (**I**), and TNF (**J**) were determined. The results are expressed as mean ± standard deviation (SD). The *p*-value was determined using one-way ANOVA followed by Tukey’s test (* *p* < 0.05, ** *p* < 0.01, *** *p* < 0.001, **** *p* < 0.0001) and (^##^
*p* < 0.01, ^####^
*p* < 0.0001). Comparison with the control group (PBS) is indicated with a (#), while comparisons between groups are marked with an asterisk. The experiment was conducted three times, and three mice were used in each repetition.

**Figure 5 vetsci-11-00034-f005:**
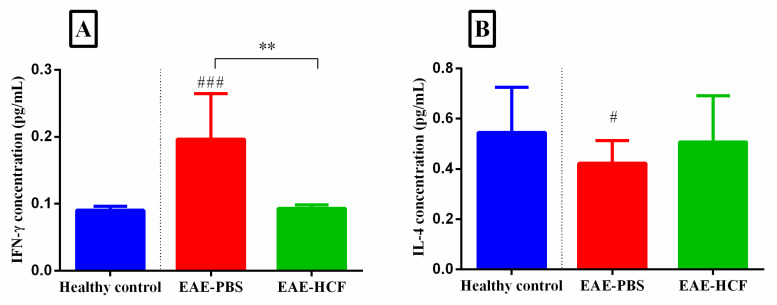
The production of IFN-γ (**A**) and IL-4 (**B**) in splenocytes was assessed using ELISA. The results are presented as the mean ±standard deviation. The *p*-value was determined by performing one-way ANOVA followed by Tukey’s test (** *p* < 0.01) and (^###^
*p* < 0.001). Comparison with the control group (PBS) is indicated with a (#). The experiment was repeated three times, with three mice used in each repetition.

## Data Availability

Data supporting the findings of this study are contained within the article.

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
