# Peer review of "Modulatory Effects of Hydatid Cyst Fluid on a Mouse Model of Experimental Autoimmune Encephalomyelitis"

_vetsci, 2024, doi:10.3390/vetsci11010034_

Round 1

Reviewer 1 Report

Comments and Suggestions for Authors

The manuscript of Hajizadeh et al, aimed to examine the effect of Cystic echinococcosis, through hydatid cyst fluid (HCF), on the disease severity in EAE. The methods are correct and well described and the results clear and well discussed.

However, I encounter a major concern with the experimental design: this consist in the treatment with HCF both before and after EAE induction. It is well known form other works that helmint infection before EAE can protect or delay EAE onset, whereas after EAE induction the effect are, in most cases, abolished. How can the author discern the role of CE infection in the prevention rather than in the therapy of MS? Other group of mice treated before or after EAE should be added to clarify the role of CE infection in the prophylactic and in the therapeutic setting of MS.  

Author Response

Dear Editor-in-Chief:                                                                          

On behalf of my co-authors, I thank you very much for giving us an opportunity to revise our manuscript, we appreciate editors and reviewers very much for their positive and constructive comments and suggestions on our manuscript entitled " Modulatory effects of hydatid cyst fluid on mouse model of experimental autoimmune encephalomyelitis" (Manuscript ID: vetsci-2780305). We have studied reviewers’ comments carefully and tried our best to revise our manuscript according to the comments. All suggestions were separately considered, addressed point-by-point and all of edited changes were highlighted. We hope that the revised form of our manuscript meets the “Veterinary Sciencesjournal and would be satisfactory for acceptance.

Reviewer 1:

Answers to comments of Reviewers

We would like to thank you very much for the very thoughtful review that helped us greatly to revise the manuscript and to improve its quality.

The manuscript of Hajizadeh et al, aimed to examine the effect of Cystic echinococcosis, through hydatid cyst fluid (HCF), on the disease severity in EAE. The methods are correct and well described and the results clear and well discussed.  However, I encounter a major concern with the experimental design: this consist in the treatment with HCF both before and after EAE induction. It is well known form other works that helmint infection before EAE can protect or delay EAE onset, whereas after EAE induction the effect are, in most cases, abolished. How can the author discern the role of CE infection in the prevention rather than in the therapy of MS? Other group of mice treated before or after EAE should be added to clarify the role of CE infection in the prophylactic and in the therapeutic setting of MS.  

Answer: Thank you for your suggestion, and we appreciate your understanding of the constraints we faced during the study. Your comment regarding the differentiation between the prevention and therapeutic effects of hydatid cyst fluid (HCF) in multiple sclerosis (MS) is indeed valid. The primary aim of our study was to investigate the potential preventive effects of HCF on the development and severity of MS. We hypothesized that HCF, derived from cystic echinococcosis (CE), could modulate the immune response and potentially delay or prevent the onset of MS. This hypothesis was based on previous research indicating that helminth infections, when acquired before MS induction, can offer protection or delay the onset of the disease. While we had originally planned to include a separate group to investigate the therapeutic effects of HCF in established MS, limitations, particularly financial constraints and time considerations, prevented us from doing so in this study. We acknowledge that this is a limitation of our work and an area for further investigation. We understand the importance of discerning the specific roles of CE infection in both the prevention and therapy of MS, and we agree that including a group treated with HCF after EAE induction would provide valuable insights into the therapeutic potential of HCF. We hope to revisit this aspect in future studies if additional resources and funding become available. We want to emphasize that our study contributes valuable evidence regarding the preventive effects of HCF on MS development and severity. However, we acknowledge the need for further research to fully understand the therapeutic potential and timing of HCF treatment in the context of established MS.

Thank you once again for your comment, and we appreciate your understanding of the limitations we encountered. Your input will help guide future studies in this field.

I deeply appreciate your kind favor and cooperation.

With the best regards, sincerely yours

Ehsan Ahmadpour, Ph.D.

Reviewer 2 Report

Comments and Suggestions for Authors

The authors have investigated the effects of E. granulosus cyst fluid administered to mice with MOG-induced EAE. They observed a shift in the immune response from pro-inflammatory Th1 to anti-inflammatory Th2, with decreased IFN-γ and increased IL-4. Also, the treatment delayed EAE onset and reduced severity, preventing weight loss and decreasing spinal cord inflammation and demyelination.

Issues to address

-The authors should better define when and where the E. granulosus cyst fluid was collected: number of animals; amount collected from each animal; period of storage. Is it possible to standardize the content of the E. granulosus cyst fluid for further experiments? Can other groups replicate the same results using different batches of hydatid fluid?

-Figure 2. Please add dispersion bars to the graphs (either S.D. or S.E.M.)

-How many biological replicates are included in the graphs from Figure 3, 4 and 5?

Comments on the Quality of English Language

Minor editing of English language required

Author Response

 Dear Editor-in-Chief:                                                                          

On behalf of my co-authors, I thank you very much for giving us an opportunity to revise our manuscript, we appreciate editors and reviewers very much for their positive and constructive comments and suggestions on our manuscript entitled " Modulatory effects of hydatid cyst fluid on mouse model of experimental autoimmune encephalomyelitis" (Manuscript ID: vetsci-2780305). We have studied reviewers’ comments carefully and tried our best to revise our manuscript according to the comments. All suggestions were separately considered, addressed point-by-point and all of edited changes were highlighted. We hope that the revised form of our manuscript meets the “Veterinary Sciencesjournal and would be satisfactory for acceptance.

Reviewer 2:

Answers to comments of Reviewers

We would like to thank you very much for the very thoughtful review that helped us greatly to revise the manuscript and to improve its quality.

The authors have investigated the effects of E. granulosus cyst fluid administered to mice with MOG-induced EAE. They observed a shift in the immune response from pro-inflammatory Th1 to anti-inflammatory Th2, with decreased IFN-γ and increased IL-4. Also, the treatment delayed EAE onset and reduced severity, preventing weight loss and decreasing spinal cord inflammation and demyelination.

Issues to address

-The authors should better define when and where the E. granulosus cyst fluid was collected: number of animals; amount collected from each animal; period of storage. Is it possible to standardize the content of the E. granulosus cyst fluid for further experiments? Can other groups replicate the same results using different batches of hydatid fluid?

Answer: Thank you for your valuable comments. To clarify the process of obtaining the cyst fluid, we initially used a sheep host to ensure the availability of viable cysts, as some cysts in cow samples are sterile and lack protoscoleces. Furthermore, in order to maintain consistency in the challenges faced by the groups being studied, samples were collected from a host with heavily infected tissue containing numerous cysts. Specifically, the liver, which exhibited multiple cysts, was chosen for examination during slaughterhouse inspections. Subsequently, the genotype of the obtained samples was determined and confirmed using molecular methods (the predominant genotype in the Azarbaijan region of Iran being G1, as indicated by our results). Following confirmation, the hydatid cyst fluid was collected and stored at -20°C. To ensure the integrity of the samples, the protein concentration was determined using the Bradford method. It is worth noting that appropriately stored proteins and antigens can remain viable for an extended period, allowing for their use in future studies without the need to prepare new samples.

-Figure 2. Please add dispersion bars to the graphs (either S.D. or S.E.M.)

Answer: Thank you for your kind attention. SD was added to graphs.

-How many biological replicates are included in the graphs from Figure 3, 4 and 5?

Answer: Thank you for your valuable comments. All experiments were conducted in triplicate. It was added in the text and highlighted.

I deeply appreciate your kind favor and cooperation.

With the best regards, sincerely yours

Ehsan Ahmadpour, Ph.D.

Reviewer 3 Report

Comments and Suggestions for Authors

The authors are exploring important question in neuroimmunology field, i.e. the influence of parasitic helminths on the CNS autoimmunity, as suggested by the hygiene hypothesis. Although the topic is of interest and the aim of the study is highly relevant, the study design has some issues that have to be addressed by the authors to improve their manuscript.

Major points

Only one experiment is performed to explore the effects of HCF on EAE. At least one additional experiment is needed to confirm the observed effect.

EAE parameters, such as cumulative, mean and max clinical scores, day of onset and disease duration should be included to present the effect of HCF on EAE.

Mice were sacrificed on day 21. Lack of some effects might be because this is late time point. It would be adequate to analyse immune cells isolated from the CNS at this time point or from the spleen at earlier time points. This should be, at least, discussed.

Why were cells isolated from the spleen treated with HCF? Why they were not re-stimulated with MOG35-55 ex vivo? Also, Figure 5 refers serum and not splenocyte cell culture supernatants.

No data about the number of samples and number of experiments in figure legends. This should be included in the revised version of the manuscript.

Statistics for comparing three groups should be ANOVA followed by adequate test, not t-test. The test used should be indicated in the figure legends.

Figure 3

B - myelin deposition or demyelinated area?

How the authors explain that there are infiltrating cells and demyelination in healthy mice? I would say that this is the baseline of their method, or? I would suggest change in the presentation.

What is the meaning of "percent/mm2". It seems that "the percent of the examined area" is more appropriate.

What is the statistic applied in Figure 3?

The authors state in the Material and methods that "The scores were decided using the following criteria: (0) no damage or absence of cellular infiltration; (1) very weak damage or <5 inflammatory cells per microscope high power field (C/HPF); (2) weak damage or 5–10 C/HPF; (3) moderate damage or 10–15 C/HPF; (4) strong damage or 141 15–20 C/HPF; (5) very strong damage or >20 C/ HPF [31]." However, they do not show data with this scale.

Explain how was HCF standardized? Was it pull from many samples? How do the authors make sure that HCF used in repeated experiments is of the same content? I do not understand why HCF is given as 50 ug, when it is liquid at the end of the isolation. Shouldn't it be in ul?

Minor

Abstract

line 24, use conditional instead of exclamation, as it is still not clear if the lack of helminth parasites is cause for autoimmunity.

TNF instead of TNF-alpha, as there is no TNF-beta anymore.

Introduction

line 49 rephrase the sentence "Industrialized countries are now experiencing an increased incidence of multiple sclerosis (MS), the one ascribed reason of reduced helminth co-infections [4,5]."

line 84, mistakes in the sentence

Material and methods

Figure 1

rephrase "EAE and HCF induction" and present this part of the scheme as timeline (avoid too many repetitions of the same text)

line 152 rephrase "The expression levels of protein genes and transcription factors"

What was the housekeeping gene used beta-actin or SDHA?

2.4 indicate the way of HCF application

The authors should consider discussing the following reference:

Mariki A, Barzin Z, Fasihi Harandi M, Karbasi Ravari K, Davoodi M, Mousavi SM, Rezakhani S, Nazeri M, Shabani M. Antigen B modulates anti-inflammatory cytokines in the EAE model of multiple sclerosis. Brain Behav. 2023 Feb;13(2):e2874. doi: 10.1002/brb3.2874. Epub 2022 Dec 29. PMID: 36582052; PMCID: PMC9927863.

Comments on the Quality of English Language

The text should be thoroughly checked for the mistakes.

Author Response

 Dear Editor-in-Chief:                                                                          

On behalf of my co-authors, I thank you very much for giving us an opportunity to revise our manuscript, we appreciate editors and reviewers very much for their positive and constructive comments and suggestions on our manuscript entitled " Modulatory effects of hydatid cyst fluid on mouse model of experimental autoimmune encephalomyelitis" (Manuscript ID: vetsci-2780305). We have studied reviewers’ comments carefully and tried our best to revise our manuscript according to the comments. All suggestions were separately considered, addressed point-by-point and all of edited changes were highlighted. We hope that the revised form of our manuscript meets the “Veterinary Sciencesjournal and would be satisfactory for acceptance.

Answers to comments of Reviewers

We would like to thank you very much for the very thoughtful review that helped us greatly to revise the manuscript and to improve its quality.

Reviewer 3:

The authors are exploring important question in neuroimmunology field, i.e. the influence of parasitic helminths on the CNS autoimmunity, as suggested by the hygiene hypothesis. Although the topic is of interest and the aim of the study is highly relevant, the study design has some issues that have to be addressed by the authors to improve their manuscript.

Major points

Only one experiment is performed to explore the effects of HCF on EAE. At least one additional experiment is needed to confirm the observed effect.

Answer: Thank you for your valuable comments. All experiments were conducted in triplicate. It was added in the text and highlighted.

EAE parameters, such as cumulative, mean and max clinical scores, day of onset and disease duration should be included to present the effect of HCF on EAE.

Answer: Thank you. It was added in the text and highlighted.

Mice were sacrificed on day 21. Lack of some effects might be because this is late time point. It would be adequate to analyse immune cells isolated from the CNS at this time point or from the spleen at earlier time points. This should be, at least, discussed.

Answer: Thank you to the esteemed reviewer. Prior to conducting this study, we thoroughly reviewed recent studies in this field. Additionally, we meticulously determined the specific duration of a small-scale (pilot) study. It should be noted that after the fourth week, the symptoms of the disease disappeared. Ultimately, we were able to identify the optimal timeframe of 21 days.

Why were cells isolated from the spleen treated with HCF? Why they were not re-stimulated with MOG35-55 ex vivo? Also, Figure 5 refers serum and not splenocyte cell culture supernatants.

Answer: Thank you for your attention. You are right, it was a typo and it has been corrected.

No data about the number of samples and number of experiments in figure legends. This should be included in the revised version of the manuscript.

Answer: Thank you for your suggestion. It was added and highlighted.

Statistics for comparing three groups should be ANOVA followed by adequate test, not t-test. The test used should be indicated in the figure legends.

Answer: We thank the reviewer for their valuable input. In accordance with the comments of the respected reviewer, all statistical analyses were revised and re-conducted.

Figure 3, B - myelin deposition or demyelinated area? How the authors explain that there are infiltrating cells and demyelination in healthy mice? I would say that this is the baseline of their method, or? I would suggest change in the presentation. What is the meaning of "percent/mm2". It seems that "the percent of the examined area" is more appropriate.

Answer: Thank you for the detailed opinion of the honorable reviewer. It was changed in the accordingly.

What is the statistic applied in Figure 3?

Answer: Thank you for your attention. As mentioned in the legend of Fig. 3, ANOVA test was used.

The authors state in the Material and methods that "The scores were decided using the following criteria: (0) no damage or absence of cellular infiltration; (1) very weak damage or <5 inflammatory cells per microscope high power field (C/HPF); (2) weak damage or 5–10 C/HPF; (3) moderate damage or 10–15 C/HPF; (4) strong damage or 141 15–20 C/HPF; (5) very strong damage or >20 C/ HPF [31]." However, they do not show data with this scale.

Answer: Thank you for your suggestion. Actually it was mentioned in Fig 2 as “Average clinical score”.

Explain how was HCF standardized? Was it pull from many samples? How do the authors make sure that HCF used in repeated experiments is of the same content? I do not understand why HCF is given as 50 ug, when it is liquid at the end of the isolation. Shouldn't it be in ul?

Answer: Thank you for your valuable comments. To clarify the process of obtaining the cyst fluid, we initially used a sheep host to ensure the availability of viable cysts, as some cysts in cow samples are sterile and lack protoscoleces. Furthermore, in order to maintain consistency in the challenges faced by the groups being studied, samples were collected from a host with heavily infected tissue containing numerous cysts. Specifically, the liver, which exhibited multiple cysts, was chosen for examination during slaughterhouse inspections. Subsequently, the genotype of the obtained samples was determined and confirmed using molecular methods (the predominant genotype in the Azarbaijan region of Iran being G1, as indicated by our results). Following confirmation, the hydatid cyst fluid was collected and stored at -20°C. To ensure the integrity of the samples, the protein concentration was determined using the Bradford method. It is worth noting that appropriately stored proteins and antigens can remain viable for an extended period, allowing for their use in future studies without the need to prepare new samples. This data was added to the text and highlighted.

Minor

Abstract

line 24, use conditional instead of exclamation, as it is still not clear if the lack of helminth parasites is cause for autoimmunity.

Answer: It was corrected.

TNF instead of TNF-alpha, as there is no TNF-beta anymore.

Answer: We appreciate for your attention. It was corrected.

Introduction

line 49 rephrase the sentence "Industrialized countries are now experiencing an increased incidence of multiple sclerosis (MS), the one ascribed reason of reduced helminth co-infections [4,5]."

Answer: It was corrected.

line 84, mistakes in the sentence

Answer: It was corrected.

Material and methods

Figure 1

Rephrase "EAE and HCF induction" and present this part of the scheme as timeline (avoid too many repetitions of the same text)

Answer: Thank you for your valuable suggestion. The timeline for EAE and HCF induction is illustrated in Figure 1, which can be found in column 2. Please refer to Figure 1 for a visual representation of the experimental timeline, including the specific days of EAE induction, HCF treat initiation and continuation, and the day of clinical assessment. Additionally, daily measurements of body weight were taken throughout the experiment.

line 152 rephrase "The expression levels of protein genes and transcription factors"

Answer: It was corrected.

What was the housekeeping gene used beta-actin or SDHA?

Answer: It was corrected.

2.4 indicate the way of HCF application

Answer: Thank you for pointing out the clarification needed regarding the method of HCF application. In section 2.4, the EAE+HCF group received HCF treatment via intraperitoneal injection. The HCF was administered at a concentration of 50μg/ml. The treatment was given 2, 4, and 6 days both before and after the EAE induction. This information has been added to section 2.4 to provide a clear description of the method of HCF application. 

The authors should consider discussing the following reference:

Mariki A, Barzin Z, Fasihi Harandi M, Karbasi Ravari K, Davoodi M, Mousavi SM, Rezakhani S, Nazeri M, Shabani M. Antigen B modulates anti-inflammatory cytokines in the EAE model of multiple sclerosis. Brain Behav. 2023 Feb;13(2):e2874. doi: 10.1002/brb3.2874. Epub 2022 Dec 29. PMID: 36582052; PMCID: PMC9927863.

Answer: Thank you for suggestion it was added in the discussion and highlighted.

I deeply appreciate your kind favor and cooperation.

With the best regards, sincerely yours

Ehsan Ahmadpour, Ph.D.

Round 2

Reviewer 1 Report

Comments and Suggestions for Authors

I thank the authors for their prompt reply. I would like to suggest that they add the primary objective of the study in the text and abstract; they should also mention in the discussion the limitation of their study as stated in the response to the reviewer.

Author Response

Dear Editor-in-Chief:                                                                          

On behalf of my co-authors, I thank you very much for giving us an opportunity to revise our manuscript, we appreciate editors and reviewers very much for their positive and constructive comments and suggestions on our manuscript entitled " Modulatory effects of hydatid cyst fluid on mouse model of experimental autoimmune encephalomyelitis" (Manuscript ID: vetsci-2780305).

Reviewer 1

Answers to comments of Reviewers

I thank the authors for their prompt reply. I would like to suggest that they add the primary objective of the study in the text and abstract; they should also mention in the discussion the limitation of their study as stated in the response to the reviewer.

We appreciate the valuable input provided by the respected referee. In response to the suggestion, we have included the primary objective of the study in both the introduction and abstract of the article. Furthermore, we have addressed the limitation mentioned in our previous response by incorporating it into the discussion section of the article. These changes have been highlighted. Thank you for bringing these important points to our attention.

I deeply appreciate your kind favor and cooperation.

With the best regards, sincerely yours

Ehsan Ahmadpour, Ph.D.

Reviewer 2 Report

Comments and Suggestions for Authors

Raised issues were addressed by the authors

Comments on the Quality of English Language

Minor editing required

Author Response

Dear Editor-in-Chief:                                                                          

On behalf of my co-authors, I thank you very much for giving us an opportunity to revise our manuscript, we appreciate editors and reviewers very much for their positive and constructive comments and suggestions on our manuscript entitled " Modulatory effects of hydatid cyst fluid on mouse model of experimental autoimmune encephalomyelitis" (Manuscript ID: vetsci-2780305).

Reviewer 2:

Answers to comments of Reviewers

Comment: Minor editing required

We would like to thank you very much for the very thoughtful review that helped us greatly to revise the manuscript and to improve its quality. In response to the valuable feedback from the esteemed reviewer, we have undertaken a comprehensive revision of the manuscript with the assistance of an expert English editor.

I deeply appreciate your kind favor and cooperation.

With the best regards, sincerely yours

Ehsan Ahmadpour, Ph.D.

Reviewer 3 Report

Comments and Suggestions for Authors

There are still some points from the original review that have to be addressed.  Although the authors claim that they have addressed them, I cannot see the appropriate changes.

Figure 3, B - myelin deposition or demyelinated area? How the authors explain that there are infiltrating cells and demyelination in healthy mice? I would say that this is the baseline of their method, or? I would suggest change in the presentation. 

The authors state in the Material and methods that "The scores were decided using the following criteria: (0) no damage or absence of cellular infiltration; (1) very weak damage or <5 inflammatory cells per microscope high power field (C/HPF); (2) weak damage or 5–10 C/HPF; (3) moderate damage or 10–15 C/HPF; (4) strong damage or 141 15–20 C/HPF; (5) very strong damage or >20 C/ HPF [31]." However, they do not show data with this scale. Fig3A should be changed in accordance with this scale or this part of material and methods should be changed to be in accordance with Fig 3A.

Figure 1

Rephrase "EAE and HCF induction" and present this part of the scheme as timeline (avoid too many repetitions of the same text). the authors did not change the title "EAE and HCF induction". It should be "EAE induction and HCF treatment" or something similar. 

Comments on the Quality of English Language

none

Author Response

Dear Editor-in-Chief:                                                                          

On behalf of my co-authors, I thank you very much for giving us an opportunity to revise our manuscript, we appreciate editors and reviewers very much for their positive and constructive comments and suggestions on our manuscript entitled " Modulatory effects of hydatid cyst fluid on mouse model of experimental autoimmune encephalomyelitis" (Manuscript ID: vetsci-2780305).

Reviewer 3:

Answers to comments of Reviewers

I cannot see the appropriate changes.

Figure 3, B - myelin deposition or demyelinated area? How the authors explain that there are infiltrating cells and demyelination in healthy mice? I would say that this is the baseline of their method, or? I would suggest change in the presentation.

The authors state in the Material and methods that "The scores were decided using the following criteria: (0) no damage or absence of cellular infiltration; (1) very weak damage or <5 inflammatory cells per microscope high power field (C/HPF); (2) weak damage or 5–10 C/HPF; (3) moderate damage or 10–15 C/HPF; (4) strong damage or 141 15–20 C/HPF; (5) very strong damage or >20 C/ HPF [31]." However, they do not show data with this scale. Fig3A should be changed in accordance with this scale or this part of material and methods should be changed to be in accordance with Fig 3A.

Answer: We would like to express our sincere gratitude for your thoughtful review, which has played a crucial role in the revision process and the overall improvement of our manuscript. After carefully considering the opinion of the respected reviewer, we recognized that we were unable to provide sufficient justification for the quantitative method employed in our study. As a result, we have made the decision to report the results solely in qualitative terms to ensure the accuracy and transparency of our findings. In light of this change, the items mentioned in your review have been subsequently removed from the manuscript. We truly appreciate your meticulous evaluation of our work, as it has significantly contributed to enhancing the quality and clarity of our research. Your feedback has been instrumental in guiding our revisions, and we are grateful for your valuable input.

Figure 1

Rephrase "EAE and HCF induction" and present this part of the scheme as timeline (avoid too many repetitions of the same text). the authors did not change the title "EAE and HCF induction". It should be "EAE induction and HCF treatment" or something similar.

Answer: Thank you for suggestion. In response to your feedback, we have made the necessary revisions to Figure 1, aligning it with your recommendations.

I deeply appreciate your kind favor and cooperation.

With the best regards, sincerely yours

Ehsan Ahmadpour, Ph.D.
